# Single-cell transcriptional networks in differentiating preadipocytes suggest drivers associated with tissue heterogeneity

Alfred K. Ramirez[1,2], Simon N. Dankel [3,4], Bashir Rastegarpanah[5], Weikang Cai[1], Ruidan Xue[1,7], Mark Crovella [5,6], Yu-Hua Tseng [1], C. Ronald Kahn [1✉] & Simon Kasif [2,6✉]

White adipose tissue plays an important role in physiological homeostasis and metabolic disease. Different fat depots have distinct metabolic and inflammatory profiles and are differentially associated with disease risk. It is unclear whether these differences are intrinsic to the pre-differentiated stage. Using single-cell RNA sequencing, a unique network methodology and a data integration technique, we predict metabolic phenotypes in differentiating cells. Single-cell RNA-seq profiles of human preadipocytes during adipogenesis in vitro identifies at least two distinct classes of subcutaneous white adipocytes. These differences in gene expression are separate from the process of browning and beiging. Using a systems biology approach, we identify a new network of zinc-finger proteins that are expressed in one class of preadipocytes and is potentially involved in regulating adipogenesis. Our findings gain a deeper understanding of both the heterogeneity of white adipocytes and their link to normal metabolism and disease.

[1] Section of Integrative Physiology and Metabolism, Joslin Diabetes Center, Harvard Medical School, Boston, MA 02215, USA. [2] Department of Biomedical Engineering, Boston University, Boston, MA 02215, USA. [3] Hormone Laboratory, Haukeland University Hospital, 5020 Bergen, Norway. [4] Mohn Nutrition Research Laboratory, Department of Clinical Science, University of Bergen, Bergen, Norway. [5] Department of Computer Science, Boston University, Boston, MA 02215, USA. [6] Graduate Program in Bioinformatics, Boston University, Boston, MA 02215, USA. [7] Deceased: Ruidan Xue. ✉email: c.ronald.kahn@joslin.harvard.edu; kasif@bu.edu

Adipose tissue is a heterogeneous organ and composed of several cell types, including mature adipocytes, preadipocytes, stem cells, endothelial cells, and various blood cells[1]. At least three distinct developmental types of adipocytes have been identified: white, brown, and beige adipocytes[2]. Different adipose depots have distinct physiological functions associated with their anatomical location and cell composition. Accumulation of visceral (intra-abdominal) white adipose tissue is associated with insulin resistance and metabolic syndrome[3], whereas accumulation of subcutaneous adipose tissue is metabolically benign and may be even associated with increased insulin sensitivity[4]. Determining the mechanisms for these phenotypic differences could lead to development of therapies for diabetes, obesity, and their associated morbidities.

The formation of new adipocytes, adipogenesis, is a well-characterized cellular model of differentiation. The process depends on several key transcriptional regulators including *PPARγ*, *CEBPβ*, and various Krüppel-like Factors (KLFs)[5,6]. Advances in technologies such as high-throughput sequencing have revealed additional transcriptional regulators ranging from zinc-finger proteins (ZFPs/ZNFs) to long non-coding RNAs[7,8]. Each of the different types of adipocytes also have distinct transcription factors, such as *ZFP423* for white adipocytes[9] and *EBF2* for brown/beige adipocytes[10], or developmental markers including *TBX15* and various homeobox genes[11,12].

A central challenging question in research of metabolic disease is the contributions of genetic, epigenetic and environmental factors. A variant of this question is whether disease risk is intrinsic to a subset of fat cells that interact with environmental stresses in disease pathogenesis. This is especially relevant since recent studies in mice have suggested developmental heterogeneity even among white adipocytes[12,13]. We attempted to address this question for human white fat using a synergistic application of several methodologies: single-cell transcriptional profiling coupled with clonal expansions in relevant tissues, network analysis and data integration using gene signatures.

Single-cell RNA sequencing is an ideal technique to profile gene expression of heterogeneous cell populations obtained from a single tissue such as blood or brain[14,15]. However, the mechanisms driving cellular heterogeneity are not well understood. We aim to develop methods that could lead to a better understanding of the potential drivers of cellular heterogeneity. We describe a network algorithm and apply it to single-cell data in reference[16] that we integrated with new RNA seq data in clonally expanded preadipocyte cell lines generated to obtain metabolic phenotypes. Such phenotyping is challenging in single-cell gene expression profiling. The algorithm and additional analysis reveals a gene network that is predicted to be associated and potentially drive adipocyte differentiation and heterogeneity.

## Results

scRNA-seq reveals at least two distinct cell populations in differentiating preadipocytes. Raw data from single-cell RNA sequencing of white adipocytes undergoing differentiation was obtained from Soumillon et al.[16]. For these experiments, primary human abdominal subcutaneous preadipocytes derived from a single donor were seeded at 100% or 80% confluency and treated with an adipogenic cocktail for either 7 days for the confluent cells or 14 days for the subconfluent cells[16]. For each day, cell cultures were trypsinized, sorted as single cells into 384-well plates by flow cytometry, RNA was extracted and sequenced. A total of ~2000 cells were analyzed for the 7-day protocol and ~6000 cells for the 14-day protocol. In our analysis, the data was first subjected to Pathway and Gene Set Overdispersion Analysis (PAGODA) to identify potential cellular heterogeneity[17]. For cells

beginning at 100% confluency, the t-SNE plots revealed at least two distinct clusters of cells at day 0, 3, and 7 of differentiation (Fig. 1a). While the two clusters tended to merge for preadipocytes at day 0 (black), by day 3 (pink), the differentiating preadipocytes separated into two distinct clusters, and this separation remained evident at day 7 (red). The experiment beginning at 80% confluency and followed over 14 days of differentiation showed a similar two-cluster separation, although in this experiment there was greater separation at the day 0, i.e., pre-confluent, cells, which then became somewhat mixed at days 1 and 2 before separating again at days 3–14 (Supplementary Fig. 1A).

PAGODA identified a set of gene signatures associated with the transcriptional heterogeneity during adipogenesis in the 7-day (Fig. 1b) and the 14-day experiment (Supplementary Fig. 1B). The genes within the highest-ranking gene sets reflected the stage of differentiation (Supplementary Fig. 1A–E). The t-SNE plots, however, showed cell clustering within each stage of differentiation. In order to determine the genes associated with clusters observed on the t-SNE plots, we performed differential gene expression between the left and right cluster for each day of analysis. First, the 7-day experiment beginning at 100% confluency in the preadipocytes (day 0), differential expression analysis between the two cell clusters produced genes primarily related to protein synthesis (Supplementary Fig. 3A). At day 3, the separation was primarily driven by genes involved in remodeling the extracellular matrix (Supplementary Fig. 3B). Finally, at day 7, in addition to the previous differences in gene expression, differential expression of genes related to metabolism appeared. These include gene sets related to oxygen consumption and protein metabolism (Fig. 1c). Genes in each of these metabolic gene sets were higher in the right cluster compared to the left, suggesting that the cells in the cluster were more metabolically active.

Whereas some differences between the 7-day and 14-day experiments were evident due to the differences in experimental design (Supplementary Fig. 4A–C), both showed a similar pattern with early differences in the two clusters being mainly in growth-related genes and late differences mainly in metabolic genes. Independently, markers of adipocytes or preadipocyte differentiation, such as *FABP4* and *PDGFRα* (Fig. 1d), were not differentially expressed between the clusters. The left cluster, as compared to the right cluster, also showed higher expression of *TBX15*, a mesodermal developmental gene, which has previously been shown to be a marker of glycolytic adipocytes in mice[11]. There was no differential expression or no detection of markers of beige or brown adipocytes, including *TMEM26* or *UCP1*, between clusters, indicating that this heterogeneity was not due to a mixture of white and beige preadipocytes.

**Integrative analysis predicts differences in glucose uptake and extracellular acidification**. To corroborate these gene expression differences and their predictive association with metabolic phenotypes, we sought to determine whether white preadipocytes from a single adipose depot clustered by metabolic phenotypes. A total of 35 clonally expanded white preadipocytes from human neck white subcutaneous tissue were generated as previously described[18]. Gene expression profiles were obtained by RNA-seq and analyzed with PAGODA. Principal component analysis revealed at least two clusters with the 35 clonally expanded cell lines (Fig. 2a). The gene sets capturing significant aspects of heterogeneity reflected differences in cell adhesion (GO:0034330), response to type I interferon (GO:0071357), and RNA processing (GO:0006396) (Fig. 2b). Some genes within these aspects linked to heterogeneity of white adipocytes in mature fat[19,20], such as

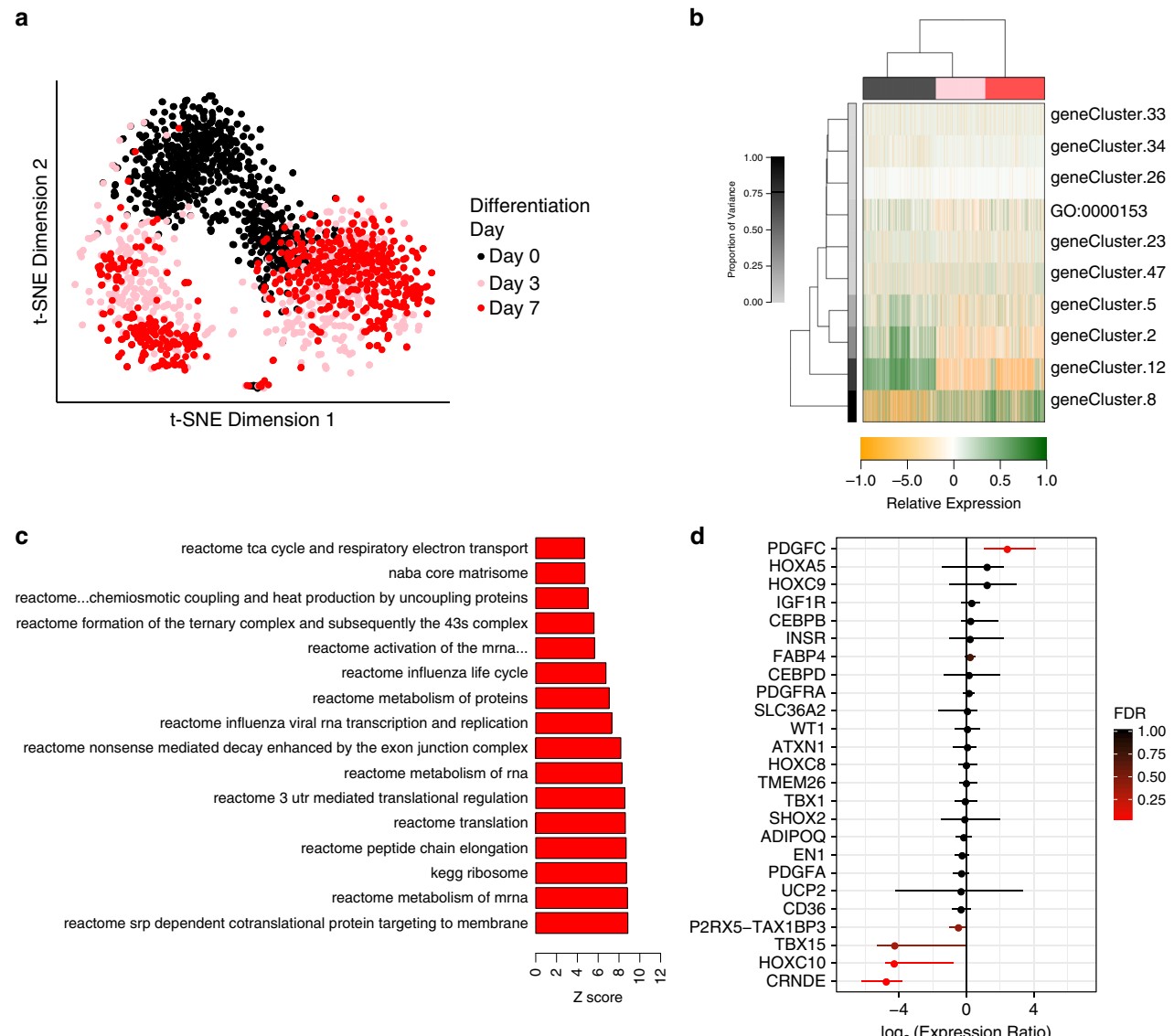

**Fig. 1 PAGODA and t-SNE reveals at least two clusters in differentiating preadipocytes. a** Single-cell RNA seq was performed on differentiating preadipocytes beginning at 80% confluency and differentiated for 7 days. PAGODA was used to determine the optimal cell clustering based on the genes driving the heterogeneity. The result was plotted by t-SNE. A total of 2092 are shown. **b** Single-cell expression profiles in were analyzed to determine the genes and pathways driving the heterogeneity. PAGODA was used to perform weighted principle component analysis on pre-defined (e.g., Gene Ontology terms) and de novo gene sets. The gene sets were scored on their significance. Correlated gene sets were coalesced in order to reduce redundancy. The heatmap of the significance gene sets shows a few de novo gene sets captured major aspects (i.e., non-redundant principal components) of heterogeneity. **c** Differential gene expression was performed between the left and right clusters for each day of differentiation in differentiating preadipocytes beginning at 100% confluency. Gene set enrichment analysis was performed on the differentially expressed genes and the top 10 up- and down-regulated pathways sorted by z-score are shown for day 7. **d** Differential gene expression was performed between the left and right clusters of day 7 adipocytes. The genes shown are previously-studied adipocyte markers for lineage or differentiation stage. For each gene, the maximum likelihood estimate and 95% confidence interval of the log2 expression ratio (right cluster over left cluster) is shown. The brown adipocyte markers *MYF5* and *UCP1* were not detected in any cells.

Cadherin-6 (*CDH6*) and Nucleophosmin (*NPM1*), showed different levels of expression in cells within the same clusters, suggesting subclusters within both clusters A and B (Fig. 2b).

Phenotypic profiling was performed on each clonal cell lines before or after differentiation to obtain profiles in extracellular acidification (ECAR), oxygen consumption (OCR), glucose uptake, and adipogenic capacity (*PPARG* expression after differentiation) (Fig. 2b). The correlation analysis was performed with phenotype profiles and the five significant gene sets identified between Cluster A and B, and showed that GO:0071357 was correlated with glucose uptake, whereas the other gene sets were not correlated to any of the measured

phenotypes (Supplementary Fig. 9). When compared to the differentially expressed genes from the day 0 clusters (Fig. 1a), we observed approximately 100 genes shared (Fig. 2c). Comparing the seven most extreme clones in the gene expression of cluster A against the seven most extreme clones of cluster B showed a difference in glucose uptake and *PPARG* expression (i.e., the capacity of differentiation into adipocytes) (Fig. 2d).

We then sought to associate heterogeneous populations identified in single-cell RNA seq with quantitative measurements in clonally expanded cell lines. Using the clonal gene expression data, phenotype-gene correlation vectors were created between every gene and each measured phenotype (Fig. 2e). These gene-

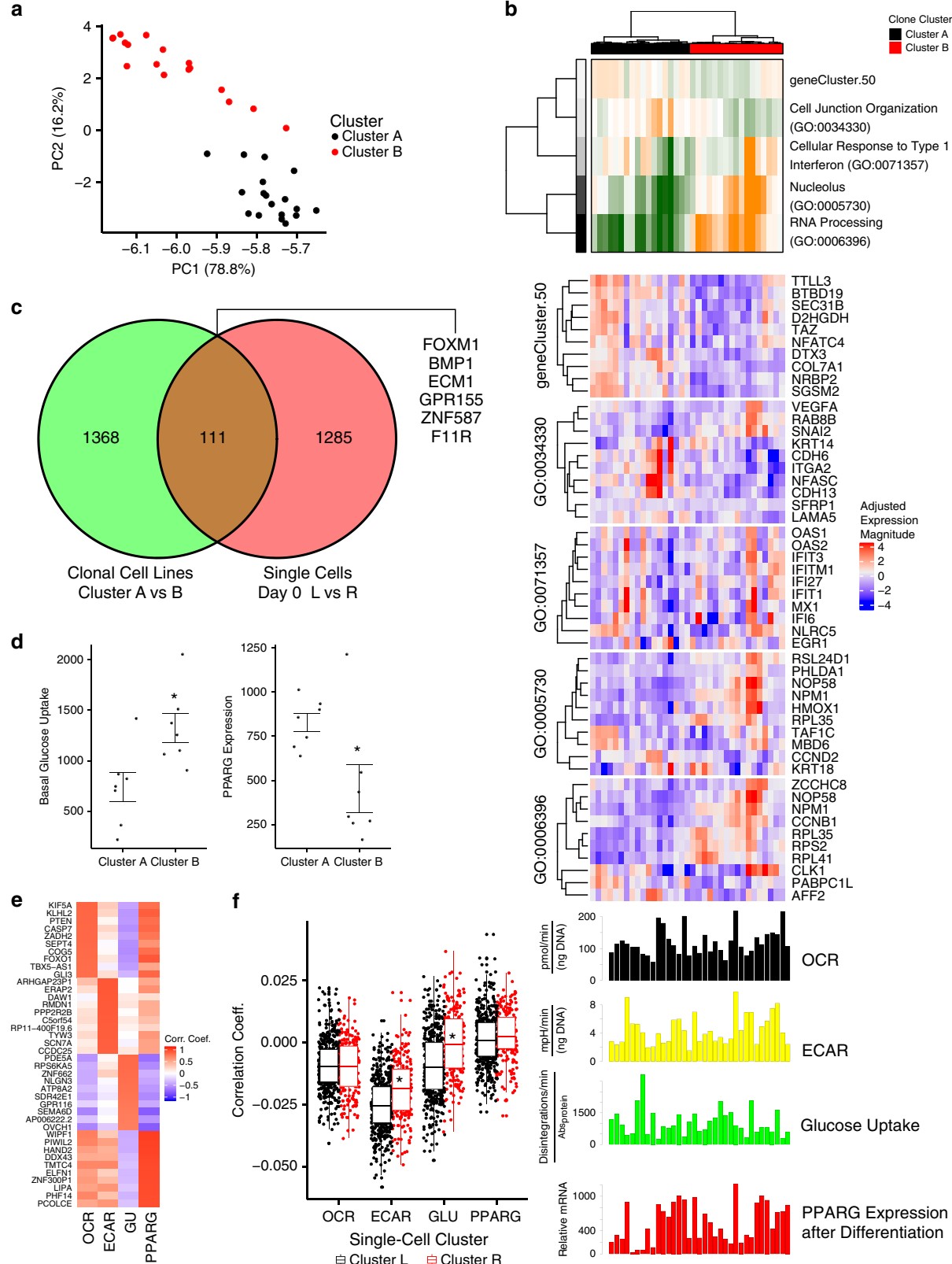

phenotype correlations were projected to the single-cell data to determine whether any gene signatures would preferentially mark the heterogeneous populations of single cells. The results indicated that the clusters showed differences in the expression of genes correlated to extracellular acidification and glucose uptake across multiple days (Fig. 2f, Supplementary Fig. 5A–H).

We then proceeded to develop a network approach to identify the potential drivers of heterogeneity in single-cell populations that revealed a network of zinc-finger proteins associated with adipogenesis-resistant cells. Combining clustering gene expression with network analysis is routinely applied to identify biological processes associated with disease-related phenotypes

**Fig. 2 Clonal preadipocyte cell lines cluster into distinct metabolic signatures.** RNA seq was performed on 35 (9 clones each from 4 subjects; 1 library failed QC) clonally expanded immortalized human neck white subcutaneous preadipocytes and analyzed with PAGODA. **a** Principal Component Analysis of subject-corrected gene expression profiles. **b** Top gene sets capturing significant aspects of transcriptional heterogeneity among the 36 clones. Each of these aspects had several genes marking potential subpopulations of clones within the clusters. Basal oxygen consumption (OCR) and basal extracellular acidification rate (ECAR) were measured in preadipocytes via the Seahorse XF Analyzer. Glucose uptake was measured with radiolabeled 2-deoxy-glucose in preadipocytes. The levels of *PPARG* were measured after 18–21 days of differentiation. **c** Venn diagram of the differentially expressed genes between clusters A and B in the clonal cell line data versus the clusters in day 0 cell in the single-cell data (see Fig. 1A). **d** The seven most extreme clones of cluster A compared against the seven most extreme clones of cluster B show differences in glucose uptake (FWER ~ 0.059) and differentiation capacity (FWER ~ 0.095). Asterisks indicate FWER < 0.1 as assessed by a one-way ANOVA followed by a Bonferroni correction ($N = 7$ clonal cells lines). Bars indicate mean ± s.e.m. **e** The four phenotypes were correlated (Spearman) to every gene in the clones to generate phenotype correlation vectors. The top ten genes for each phenotype are shown in a heatmap. **f** The four phenotype correlation vectors (oxygen consumption, extracellular acidification, glucose uptake, and *PPARG* expression after differentiation) were then correlated (Spearman) to the single-cell gene expression profiles for each cluster. The distribution of correlation coefficients was compared between the left and right cluster for each day. Boxplots are centered on the median, the interquartile range (IQR) spans the 25–75% percentile, and the whiskers extend to 1.5 times the IQR above the 75% percentile (maximum) and below the 25% percentile (minimum). Points indicate correlation coefficients of individual cells. Asterisks indicate FWER < 0.05 as assessed by an ANOVA followed by a Bonferroni correction.

in different cells types[21–23]. We developed an algorithm to detect a network of highly-activated genes that potentially code for protein complexes (protein–protein interaction (PPI) networks) in single-cell data (Fig. 3a). Our method identified at least three distinct protein networks: (1) the ubiquitin-proteasome complex (residual error = 2583), (2) the ribosome (residual error = 2587), and (3) an uncharacterized protein network (residual error = 2593). As the two higher-ranking networks captured known biology in adipogenesis, we focused on the module consisting of 30 connected genes in the PPI (Fig. 3b)[24]. These genes were enriched for proteins containing KRAB domains and C2H2 zinc-finger regions (Fig. 3c).

This module had a small tightly-connected subnetwork consisting of largely-uncharacterized zinc-finger proteins (ZNFs). Although the target genes of these ZNFs are not known, one of the highest-scoring motifs in the promoters of these genes was Interferon Regulator Factor 1 (*IRF1*) (Supplementary Fig. 7), a transcription factor which has been previously identified as a negative regulator of adipogenesis[25]. Analysis of the top 100 cells with the detected module over the time-course of adipogenesis revealed that expression of several of these ZNFs (*ZNF264*, *ZNF490*, *ZNF587*, and *ZNF714*) decreased as the level of expression of FABP4, a marker of adipocyte differentiation, increased (Fig. 3d). Interestingly, the ZNF cluster did not significantly vary in the gene expression profiles of preadipocyte cultures across the time course of preadipocyte differentiation into adipocytes (Fig. 3e). This ZNF module is only detected at the single-cell level and not at the multi-cellular population level, suggesting the module is found in a subpopulation of pre-adipocytes and not in progenitor cells of other mesodermal fates (Supplementary Fig. 8A–C).

## Discussion

Understanding the heterogeneity of white adipocytes is an important question in adipocyte biology since it is known that different patterns of fat distribution are linked to insulin resistance, different metabolic states, and the risk of diseases like type 2 diabetes and metabolic syndrome[3,26]. However, this type of analysis has been hindered by technological challenges (e.g., sorting of large and fragile adipocytes) and reliance on predefined markers, and is especially difficult to apply to human fat tissue. Here, we have addressed the question by unbiased analysis of single-cell RNA-seq of human subcutaneous preadipocytes undergoing differentiation into white adipocytes. The results show the presence of at least two distinct clusters of cells in both the preadipocyte and adipocyte stages. Differential gene expression reveals that the early differentiation differences are due to

genes related to cell cycle, protein synthesis, and growth-associated pathways. Late-stage differences are primarily due to genes related to glycolytic metabolism, which was also confirmed by phenotypic profiling in clonally-derived white preadipocyte cell lines. Lastly, a subset of cells within one of the clusters is marked by the activation of a set of zinc-finger proteins, alluding to additional potential subsets of preadipocytes including a pool of adipocyte precursors resistant to adipogenesis.

Previous research in mouse models have identified subpopulations of preadipocytes marked by the expression of certain genes such as *Sca1* and *Myf5* (refs [13,27]). It was also shown that different fat depots had different proportions of these *Sca1+* or *Myf5+* subpopulations. For example, preadipocytes derived from a *Myf5+* lineage comprised 9% of perigonadal white adipose tissue and as much as 50% of anterior subcutaneous white adipose tissue[28]. Additional studies found that knocking out PTEN in Myf5+ cells resulted in lipodystrophy[29], suggesting that these different preadipocyte populations have a functional importance and could serve as important therapeutic targets. Historically, research into human white adipose heterogeneity was limited to stromal vascular cells sorted by pre-defined markers[30,31]. Recently, there have been several efforts to utilize single-cell sequencing to determine the cell types in human white adipose tissue[32–35]. For example, Schwalie et al.[32] identified a group of anti-adipogenic stromal cells, marked by *CD142* expression, which they termed Aregs. Consistent with this, our data suggest *CD142* (gene *F3*) may be differentially expressed between the day 0 clusters in the two separate differentiation experiments (FDR ~0.44 and ~0.20). It is worth noting that these studies, as well as ours, have thus far been limited to preadipocytes and adipose tissue derived from healthy humans. The single-cell heterogeneity of adipose in diseased states, such as diabetes and metabolic syndrome, could reveal new mechanistic insights.

In adipose tissue, one of the primary challenges is to determine the transcriptional programs that give rise to different cells, potentially controlled by uncharacterized transcription factors such as zinc-finger proteins. Zinc-finger proteins are one of the largest families of DNA-binding proteins[36]. The zinc-finger proteins *ZNF264* and *ZNF490* have been shown to be DNA-binding[37], and *ZFP423* has been shown to be a marker of pre-adipocytes in the mouse[38]. In addition, the expression of *ZNF714* is higher in visceral adipose tissue from insulin-resistant obese populations compared to insulin-sensitive obese populations[39]. In the context of developmental biology, transcriptional and cellular heterogeneity have been shown to increase tissue robustness as a response mechanism to different perturbations[40–42]. In adipose tissue, the robustness is needed to maintain

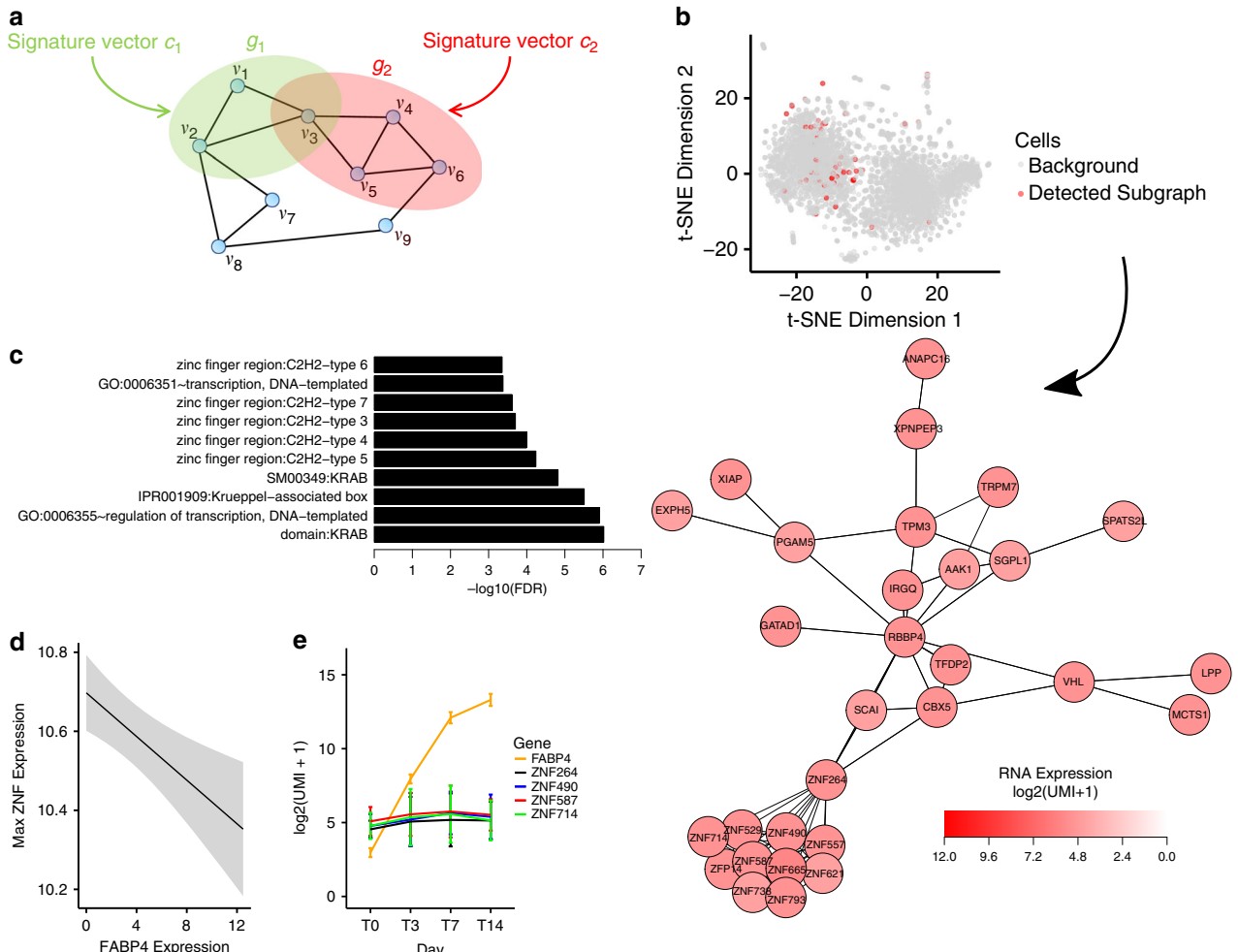

**Fig. 3 Subnetwork detection reveals a cluster of ZNFs in adipogenesis. a** An overview of the subnetwork detection algorithm where the expression of each gene can be described as the sum of underlying biological networks. **b** The subnetwork detection algorithm was used to reveal a set of cells primarily found in the left cluster. The detected subnetwork had 30 connected genes. A total of 4319 cells are shown. **c** Pathway enrichment revealed these 80 genes were enriched in KRAB domains and C2H2 zinc-fingers. **d** The mean of the max expression of the ZNFs (*ZNF264, ZNF490, ZNF587,* and *ZNF714*) in the detected cells showed a negative correlation to the adipogenic marker *FABP4*. Line indicates mean expression, gray shaded area indicates 95% confidence interval. **e** RNA-seq of whole preadipocyte cultures during differentiation show that the ZNFs do not decrease over time. Bars indicate mean ± s.e.m. $N = 2$ independent experiments.

energy homeostasis in response to perturbations such as diet, frequency of eating, and many other factors. Earlier studies have made observations consistent with at least two types of pre-adipocytes defined by several differences including adipogenic capacity, replicative capacity, and apoptosis susceptibility[13,43,44]. Thus, these ZNFs may be involved in the transcriptional regulation of white adipocyte differentiation contributing to tissue heterogeneity and robustness.

Previous work in systems biology developed numerous methods to model a biological system as a linear system[45,46]. In the context of gene expression, this framework intuitively implies that each gene in a system is a gate whose output can be modeled as linear combination of outputs of other gates with some error. While solving such systems is relatively tractable, the noise, complexity and lack of linearity in biological systems makes these efforts challenging. Here, we introduce a useful technical insight to integrate biological intuitions with computational modeling. Instead of assuming the entire system is linear, we assume the system can be decomposed into connected network modules where the system can be approximated by a low dimensional

linear factorization of the union of these network modules. The possibility of such factorization without the constraint of connected PPI modules has been previously suggested in other biological systems[47]. The network constraint adds significant technical challenge to obtaining such a model. The advent of modern omics, single-cell technologies and machine learning opens the door to methods utilizing integrative algebraic network decompositions with numerous applications in systems biology.

Our work provides preliminary support for the hypothesis that heterogeneity might be coded for early in adipogenesis. Since heterogeneity of mature adipocytes has been associated with metabolic disease risk, we can speculatively hypothesize that metabolic disease might be intricately tied to early and specific differentiation trajectories. These developmental trajectories might be affected by genetics. Alternatively, pre-metabolic disease states via yet to be fully understood signaling or communication mechanisms may modify transcriptional control in preadipocytes, driving increase in cell populations with higher disease risk. Thus, this research direction may lead to therapeutic directions reprogramming specific differentiation paths.

In summary, we have demonstrated a synergistic combination of three systems biology frameworks (single-cell transcriptional profiling, integration and joint analysis with clonal gene expression data and network modeling). We have provided evidence of the presence of at least two types of human white subcutaneous adipocytes. These adipocytes show large differences in metabolic gene sets and are not accounted for by differences in brown or beige fat. We have also shown that a subset of preadipocytes are marked by the expression of a ZNF network, which may be involved in regulating white adipocyte differentiation. Changes in the balance of these types of white adipocytes may have important physiological consequences in adipose tissue and whole-body metabolism. Dissecting intrinsic cellular heterogeneity in white pre-adipocytes and adipose tissue may significantly impact our understanding of the role of differentiation programs, genetics and environment in metabolic disease risk.

## Methods

**Base-level analysis of single-cell RNA-seq.** Single-cell RNA seq profiles of abdominal subcutaneous human preadipocytes undergoing adipogenesis were obtained as previously described[16]. In these data, the primary human preadipocytes were taken from commercially-available lipoaspirates (single donor; Life Technologies) sorted by fluorescence-activated cell sorting (FACS) based on the positive selection (i.e., express the proteins) for CD29, CD44, CD73, CD90, CD105, CD16, and negative selection (i.e., do not express) against CD14, CD31, CD45, Lin1. The cells were maintained in 2% reduced serum growth media for up to 3 passages (MesnPro, Life Technologies). For differentiation, cells were grown to either 80% or 100% confluency and then incubated with differentiation media (StemPro adipogenesis differentiation media, Life Technologies) for 7 or 14 days in two separate experiments as described[16]. For single-cell sorting, culture plates were treated with trypsin, and the released cells pelleted by centrifugation at 1000 rpm for 5 min. Pellets were resuspended with Dulbecco's phosphate-buffered saline (DPBS), stained with Hoechst 33342, and individual cells FACS sorted using the above markers into prepared 384-well plates. Library generation and RNA-sequencing was performed as described[16].

For data analysis, we filtered out cells containing less than 1000 detected genes. On average, each cell contained ~1600 detected genes. We used PAGODA[17] to analyze the expression profiles in experiment 'D1' (differentiation beginning at 80% confluency, 2092 cells after QC) and experiment 'D3' (differentiation beginning at 100% confluency, 4319 cells after QC) (GSE53638). PAGODA finds non-redundant genes sets who first principal components (i.e., aspects) are overdispersed. Significant aspects were chosen by FDR < = 0.05. Single-cell differential gene expression was performed with the package SCDE[48]. As the t-SNE plots revealed two distinct clusters, we performed differential gene expression between the left and right cluster in each day and in each separate experiment.

**Cell culture and adipogenesis.** No specific human studies approval was needed for the current study as the human cell lines used in this study were existing biobank specimens that had been collected under an approved human studies protocol for the research previously published in[18]. Human preadipocyte cell lines were obtained from superficial fat depots in the neck of patients undergoing cervical spine surgery and immortalized with stable transfection of human telomerase (TERT) as previously described[18,49]. Cells were passaged at 80% confluency and maintained in growth media (DMEM-H, 10% FBS, 1% penicillin-streptomycin). For differentiation, cells were grown to 100% confluency and incubated with pre-induction media (DMEM-H, 2% FBS, 1% Pen-Strep, 0.2% normocin, 500 nM insulin, 2 nM T3, 1 μM rosiglitazone) for 6 days, followed by the induction media (DMEM-H, 2% FBS, 1% Pen-Strep, 0.2% normocin, 500 nM insulin, 500 μM IBMX, 2 nM T3, 1 μM rosiglitazone, 33 μM biotin, 17 μM pantothenate, 100 nM dexamethasone, and 30 μM indomethacin) for 7–14 days.

**Total RNA isolation and RNA-seq of clonal preadipocytes.** Clonal human neck-derived subcutaneous preadipocyte cell lines were maintained in culture and grown to 100% confluency before RNA harvest. Total RNA was harvested and purified with Trizol reagent. RNA quality was measured on the Agilent 2100 Bioanalyzer to ensure RIN values were greater than 8.0. To construct libraries for sequencing, we used the NEBNext Ultra Directional Library Prep Kit and Poly-A selection kit (New England Biolabs). Library enrichment and multiplexing was performed using the NEBNext High-Fidelity PCR Master Mix (14 cycles of PCR) and NEBNext Multiplex Oligos. The cDNA libraries were multiplexed on the Illumina HiSeq 2000 and 2500 to generate raw reads and deposited on GEO (GSE128253). Raw 50 bp paired-end reads were aligned with STAR (v2.3.0e)[50] to the human genome build hg19 and annotation file gencode v19. Raw counts were extracted using HT-seq (v0.5.4). Analyses were carried out in R and various packages including DESeq2 (ref.[51]) and SCDE[48]. Subject-specific gene expression profiles were normalized

with SCDE. To estimate the number of clusters in the subjected-corrected expression profiles, we performed k-medoids clustering and selected the $k$ that maximized the average silhouette width ($k = 2$–10). To examine potential phenotypically-distinct subclusters within these two large clusters defined by gene expression, we performed hierarchical clustering and cut the tree to yield five clusters with average cluster size of 7 cell lines.

**Extracellular flux profiling.** Oxygen consumption rate (OCR) and extracellular acidification rate (ECAR) were measured using a Seahorse XF24 (Seahorse Bioscience). Cells were counted with a Cellometer® Vision (Nexcelom Bioscience) and seeded in 24-well Seahorse Cell Culture Microplates (60,000 cells/well). Each clonal cell line was seeded in duplicate or triplicate (one 24-well plate per subject). 1.5 h after the seeding in 100 μl medium (4.5 g/L glucose DMEM, 10% FBS, 1% penicillin/streptomycin), an additional 150 μl medium were added to ensure optimal distribution of cells. The metabolic profiles of the cells were analyzed by Seahorse either 24 h later or after a 21-day differentiation protocol as described above. 2 h prior to measurements, 150 μl of culture medium was removed from each well. The cells were washed once with 900 μl of XF Assay Medium (1 g/L glucose, 2 mM L-glutamine, 2 mM Na pyruvate), followed by the addition of XF Assay Medium to a final volume of 500 μl and incubation at 37 °C with no $CO_2$. Basal oxygen consumption and extracellular acidification were measured for 30 min. DNA was extracted by washing cells with PBS, incubating in 100 μl 50 mM NaOH for 30 min, heating at 96 °C for 5 min, and adding 25 μl 1 M Tris pH 6.5. DNA concentrations were measured by NanoDrop. The OCR and ECAR values of preadipocytes were normalized to the DNA contents in the corresponding well. OCR and ECAR in fully differentiated adipocytes were normalized to intracellular Oil Red O. Data were processed and exported with Seahorse Wave (2.3).

Basal glucose uptake was measured by intracellular incorporation of radioactive-labeled $^{3}$H-deoxy glucose. Cells were seeded in 12-well plates (50,000 cells/well, duplicates) and cultured in DMEM containing 4.5 g/L glucose, 10% FBS and 1% penicillin/streptomycin for 48 h, before washing cells twice with PBS and changing to 1 g/L glucose DMEM with 5% FBS for 16 h. Cells were then washed once with PBS and once with KRH buffer (7.4 pH, 10 mM HEPES, 25 mM Glucose, 1 mM $MgCl_2$, 1.8 mM $CaCl_2$, 4 mM KCl, 116 mM NaCl) supplemented with 2.5 mM pyruvate and 0.5% fatty acid-free BSA, before incubation at 37 °C for about 3 h in 1 ml of the glucose-free KRH. The cells were washed twice with PBS, incubated in 0.5 ml KRH/BSA containing 100 μM 2-deoxy-glucose and 0.5 μCi for 2 min, and placed on ice to terminate glucose uptake. 400 μl of each sample were transferred to scintillation tubes and 4 ml CytoScint scintillation fluid were added before rigorous shaking and measurement in a Beckman LS6500 scintillation counter. Disintegrations per minute (DPM) were normalized to protein contents of each well measured by BCA Protein Assay Kit (Pierce).

**QPCR analysis.** For qPCR analysis, cDNA was synthesized from 500 ng RNA using the High Capacity Reverse Transcription Kit (Applied Biosystems). In 20 μl reactions, 2 μl RT-buffer, 0.8 μl dNTP mix, 2 μl random primers, and 1 μl Multi-Scribe RT were mixed together with RNA and nuclease-free water. The samples were incubated at 25 °C for 10 min, 37 °C for 2 h and 85 °C for 5 min. The cDNA was diluted 1:20 in nuclease-free water. qPCR was performed using the CFX384 Real-Time System (BioRad). A master mix was prepared containing 5 μl 2X SYBR Green, 0.1 μl forward primer, 0.1 μl reverse primer, and 2.3 μl nuclease-free water per sample to which 2.5 μl cDNA template was added in a 384-well plate. The reaction was run by the following program: 95 °C for 3 min, then 40 cycles of 95 °C for 10 s and 60 °C for 45 s. Melting curve analysis (increments of 0.5 °C from 60 °C to 95 °C) confirmed specificity of the primers. Mean target mRNA levels based on technical triplicates were calculated by the ΔΔCT method and normalized to the mRNA level of HPRT. Primers for PPARG (Forward: 5′- GAA AGC GAT TCC TTC ACT GAT-3′; Reverse: 5′-TCA AAG GAG TGG GAG TGG TC-3′) and HPRT (Forward 5′- TGA AAA GGA CCC CAC GAA G-3′; reverse: 5′- AAG CAG ATG GCC ACA GAA CTA G-3′) were ordered from IDT.

**Subnetwork detection by integrating the protein–protein interactome and module detection.** In this paper, we deploy a network algorithm to analyze single-cell RNA seq data in differentiating preadipocytes. We believe that the integration of network analysis and single-cell RNA seq is a very promising direction for understanding both modularity and heterogeneity of gene expression in complex biological processes. Our hypothesis is that the gene expression state of the cell is, in part, influenced by an underlying biological network defined by the interactivity between proteins. This hypothesis is consistent with a number of publications in system biology[21,22,52]. This framework is different from using a regulatory network that captures protein-DNA interactions[53,54]. Our underlying network is a wiring diagram where the nodes are proteins and the edges are literature-reported experimentally-observed interactions. It should be noted that we assume these literature-reported interactions would be present in all cells expressing the proteins, i.e., they are tissue independent[24].

Given our hypothesis, each gene expression profile of a single-cell defines a network snapshot. Previous publications using a regulatory network framework for modeling gene expression[54,55] attempt to recover the causal regulators of gene

expression[56–58]. In contrast, we aim to produce a relatively simple representation capturing the activity of protein complexes, transcriptional modules or signaling pathways in different cells. In the simplest case, the goal is to explain the system behavior by predicting up-regulated and down-regulated modules and their activity in heterogeneous cell populations during adipogenesis.

The input to our algorithm is:

(a) A matrix containing the single-cell expression profiles in experiment 'D1' was first transformed to log scale (log2(UMI + 1) and then transformed to z-scores.
(b) The input protein–protein interaction network was the union of all interactions in the network database Pathway Commons v5 (network named 'Pathway Commons.5.All.EXTENDED_BINARY_SIF')[24].

Integrating the two data sources, we can interpret the $N$ single-cell gene expression profiles as a set of $N$ network snapshots in different time points of adipocyte differentiation. Each node (protein in the network) is associated with a score that corresponds to the differences in gene expression of the corresponding gene in this cell as compared to distribution of all values of this gene in the data. In this analysis, the edges are unweighted, although the method can be generalized to edges associated with confidences as describe in STRING and related databases[59,60].

The single-cell network snapshots can be described as a matrix of row vectors $Y_1$, $Y_2$, …, $Y_N$ where each $Y_i$ corresponds to a gene expression profile of a single cell. This yields a matrix where the columns are nodes in the network and the rows are network snapshots (gene expression profiles of cells). The goal of the algorithm is to represent the behavior of the network during differentiation as the activity profiles of relatively small connected components of the network. Within each network component, the activity is captured by a relatively *simple explanation*. For example, the activity of the network could be explained by a connected component in which the expression of the genes is relatively high on average, which could define a module associated or driving a state of differentiation. Alternatively, we may be able to identify a protein complex that consists of subunits of the same protein, such as the ribosome (a protein complex) and its many subunits (the proteins *RPS2*, *RPS3*, etc).

Unlike standard statistical analysis that relies on methods such as piecewise multiple linear regression or clustering we add two relevant biological constraints:

(a) The genes in each component are connected in the PPI network (module).
(b) The activity of each genes across the cells can be mathematically described as a linear function of multiple connected networks that the gene appears in. For example, a single gene can be controlled by two or more networks: one or more of which could be activating and the other(s) repressing.

This approach extends the class of complex systems we can explain relatively succinctly with connected modules using matrix algebra. Pathway enrichment of the resulting networks was performed with DAVID[61]. Further detail is provided in Supplementary Methods and code is available at https://github.com/mcrovella/CG-Demo.

**Analysis of ZNF expression in a single-cell RNA seq dataset of adipogenesis, chondrogenesis, and osteogenesis**. We analyzed a single-cell RNA seq data set containing profiles of the differentiation of fibroblasts into adipocytes, chondrocytes, and osteocytes (GSE37521)[62]. The normalized, pre-processed RNA-seq data were downloaded from GEO and analyzed in R via the packages GEOquery[63], Limma[64], and Biobase[65]. For comparing the ZNF expression to FABP4, a marker of adipogenesis, for each cell, the max of *ZNF264*, *ZNF490*, *ZNF587*, or *ZNF714* was used.

**Motif analysis**. Motif analysis of the genes was performed with the R package motifmatchr[66]. The promoter region was defined as the region between 2000 bp of the transcription start site in the human genome build hg19. A significant match was set to $p < 1e-5$. The transcription factor database was from JASPAR via the R package JASPAR2018.

**Statistical analyses**. Statistical tests were performed as indicated in the figure captions. Briefly, single-cell differential gene expression was performed as described in the R package SCDE[48]. Statistical significance was assessed by an unpaired, two-tailed t-test or an ANOVA followed by Bonferroni correction. All statistical analyses were performed in R.

**Reporting summary**. Further information on research design is available in the Nature Research Reporting Summary linked to this article.

## Data availability

The RNA-seq data on clonally expanded human neck-derived white preadipocytes are available at GSE128253. The single-cell RNA-seq data of white preadipocytes during in vitro differentiation are available at GSE53638. The single-cell RNA-seq data of adipose-derived stem cells during differentiation into three different mesenchymal lineages are available at GSE37521.

## Code availability

The code to run CG decomposition is available at https://github.com/mcrovella/CG-Demo.

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

## Acknowledgements

The authors would like to thank Evimaria Terzi for contributions to the development of the network algorithm; Drs Jonathan M. Dreyfuss and Hui Pan from Joslin Diabetes Center DRC Genomics and Bioinformatics Core for assistance and advice regarding data analysis; Mary Ellen Fitzpatrick for her assistance with the Green Cluster Computing Environment; and Guoxiao (Grace) Wang for her assistance with CRISPR knockouts. A. K.R. was supported by NIH grant T32 DK007260-37. B.R. and M.C. were supported by NSF IIS-1421759 and NSF CNS-1618207. This work was supported in part by NIH grant (R01DK082659 to C.R.K.). We also received support from the Joslin NIH-funded DRC (P30DK036836).

## Author contributions

A.K.R., C.R.K. and S.K. designed and provided the overall oversight of the study. A.K.R., S.N.D., B.R., M.C., W.C., C.R.K. and S.K. wrote the paper. A.K.R., R.X. and S.N.D. performed the experiments. A.K.R. analyzed gene expression data. A.K.R., B.R., M.C. and S.K. contributed to the design of the single-cell analysis methods. A.K.R., Y.H.T., C.R.K. and S.K. analyzed and discussed the data and biological results. B.R. and M.C. wrote the code for the subnetwork detection. This work was supported in part by NIH grants P30DK036836, R01DK0835659 and R37031036.

## Competing interests

The authors declare no competing interests.
