## [Peer Review File · Nature Communications]

Reviewers' comments:

Reviewer #1 (Remarks to the Author):

The authors investigate the transcriptional and functional heterogeneity of human pre-adipocytes before, during and after adipogenic differentiation using different single cell-based analysis approaches and tools. They show that distinct subpopulations of cells exist in the subcutaneous white adipose tissue that may be associated with defined metabolic phenotypes and propose that this translates cellular heterogeneity into states of metabolic disease susceptibility. Lastly, they introduce a previously unrecognized cluster of zinc finger transcription factors that may be involved in the regulation of cell-type specific metabolic function. Overall, this study is exiting and makes use of novel technologies and integrative approaches using single cell analysis and characterization of clonally expanded pre-adipocytes. There are a few items that the authors should consider in a revised manuscript, as I am not entirely sure that all statements made in the manuscript are fully supported by the data.

Major:

- A concern is that the authors describe their results on cell heterogeneity as potentially related to disease risk yet an analysis of cells derived from metabolic disease patients is not part of the analysis. With the present data it is difficult to determine whether the heterogeneity described would indeed be related to pathology. Additional experiments with metabolically impaired patients would be extensive so this should at least be discussed more clearly. The dataset presented in this study is in itself highly interesting and of significant value to the community, even without making in more than it is.
- I would like to see a more detailed description of the single cell analysis beyond the mere mention of the analysis package that was used. Some important questions specific to the layout of the experiment described here need to be addressed, for instance how potential batch effects of experiments done on different days were tested and corrected for. Similarly, please provide a more detailed set of information on how library quality was tested and describe parameters for excluding cells which is likely in any scRNA-Seq approach. Some basic information seems to be missing, such as number of single cell libraries included/excluded in the analyses shown in main and suppl. figures.
- It would be helpful if the authors could determine in more detail whether there are any similarities between the two clusters shown in figure 1 and the clusters that potentially exist in figure 2A. Specifically, I would like to see the extent of overlaps on the level of individual differentially expressed genes between these two experiments. What is the percentage of differentially expressed genes that overlap between the two clusters? Please provide some type of Venn diagram and also lists of genes that do overlap.
- I am not sure I understand the data shown in figure S8: The distribution of gene expression across time points and differentiation strategies seems rather random and does not seem to reflect the description of the data in the figure legend or the text. This may be a misunderstanding, but it would then be helpful to clarify this. For instance, I do not see evidence for downregulation of the ZNF cluster during osteogenesis.
- The data shown in S9 are not sufficient to prove enhanced or accelerated adipogenesis, especially since the data shown for the later time point somewhat contradict this conclusion. The authors should therefore conduct a comprehensive characterization of adipogenesis and metabolic function in these knockout cell lines to support their conclusion of a novel mechanism of ZNF proteins that accelerate adipogenesis. In particular, a time course of gene expression analysis and lipid accumulation would be essential to prove the assumption of accelerated adipogenesis.

Minor

- ATXN1, a protein localized to the nucleus, is not the human homolog of murine Sca1 (a cell surface protein), it does not exist in humans.
- Lines 212-214: I am not sure I would characterize the identification of an adipogenesis-accelerating gene as "novel", since there is ample evidence for quite a few genes or mechanisms

or reagents that enhance / accelerate adipogenesis in the literature.

- A recent study introduced the concept of regulatory cells in adipose tissues, the A-regs, which express CD142 as surface protein (Schwalie et al, Nature 2018). Are there any links or similarities between that study and the clusters identified here?

- Lines 246-250: I am not sure I can follow the reasoning of this passage: If there is a small pool of cells that are resistant to adipogenesis, how would they be useful in a context of transient increased nutrient supply or unknown metabolic demands? Especially if there are plenty of other, non-resistant progenitors to produce adipocytes? The avoidance of adipogenesis that the authors consider useful would contradict the physiological imperative to store nutrients effectively. Potentially related to previous comment.

- Lines 255-259: The data presented here are very exciting and could contribute to further the field significantly. I am not sure, however, that this concept is novel in the sense that it challenges "traditional thinking in disease biology". The authors may want to reconsider such provocative statements and be more realistic, especially given that they do not explore their hypothesis experimentally and in relation to a disease and that the evidence in relation to the ZNFs they introduce has not been extensively evaluated by experiments.

- Portions of the discussion seem rather unrelated to the results described in the paper. The authors could consider revising some paragraphs (for instance: lines 260-264) to make the discussion more succinct.

Reviewer #2 (Remarks to the Author):

Summary and General Comments:

This manuscript describes the use of an unsupervised data generation tool, single cell RNA-seq, the development of an algorithm integrating this data with existing structured biological knowledge to generate hypotheses, and the experimental validation of these hypotheses. The study uses single cell RNA-seq (scRNA-seq) on human preadipocytes from a previous unpublished study to find two clusters of preadipocytes that did not correspond with known subtypes of these cells. These observations are supported by additional RNA-Seq and metabolic profiling is done on clonally-expanded preadipocytes. The single cell expression data is then integrated with PPI networks to identify a novel network associated with one subset of cells, and CRISPR knockdown of some of these genes supports their involvement in the differentiation of these cells. I concur with the abstract that this represents an impressive example of the use of an unsupervised experimental approach with algorithmic innovations to describe a novel mechanism in a biological system. Each step in this process requires extensive knowledge and trial and error, and it is impressive to see all steps included in one study. I recommend acceptance of this manuscript with several small changes.

Major Comments:

Some analysis of the statistical significance of the modules discovered by the network method should be included. The way this study is described, it seems like the zinc finger network may have been the clear choice for follow-up experiments, but in other studies where multiple networks could be found, an understanding of the significance of the modules would be important for prioritizing follow up. In particular,

- There should be more quantitative motivation for focusing on the ZNF submodule specifically. While there is a highly connected submodule of ZNFs, that reason alone does not necessarily mean that the ZNFs are any more functionally important than the other genes in the module. The authors do link the ZNFs to adipogenesis through IRF1, but that is only the fifth most enriched TF in figure S7 (and that TF ranking does not come with any measure of statistical significance) -- are any of the other TFs involved in adipogenesis, and what happens if you try to perturb IRF1? In the current form of the manuscript, focusing on ZNFs for the perturbation experiments (even if they

were successful) seems a little arbitrary.

- The authors really need to make clear exactly what kind of biological models and replicates are being used for both scRNA-seq and bulk experiments. How many donors? Were all cells immortalized?

- Figure S9 in the supplement is the key biological finding and should be moved to the main text. Again, the authors should also make clear what kind of replication is happening; the current description of "3 different transfections" is ambiguous. Are these all in the same cell line or different cell lines?

Minor Comments:

- Further description of the single cell RNA-seq data analysis is necessary – ex. How many cells were sequenced vs filtered out during QC? How many genes were found in each cell, on average?

- Why are only days 0, 3, and 7 used in this study when the original study included additional time points

- Our understanding from the previous study is that this single cell RNA-Seq data is from a single human donor. If this is true, this should be pointed out in the manuscript as it limits the generalizability of the biological findings.

- Figure 2b would be more readable if the human-readable names of the GO terms were used as labels in addition to or instead of the GO accession numbers.

- Lines 161-162 describes an analysis of correlation with a phenotype. Does this refer to correlation of the individual genes in these GO lists or some sort of score on the GO terms? How was the significance of this correlation determined since different GO terms contain different numbers of genes?

- Throughout the manuscript (ex. caption to figure 2, and line 151), the word "aspects" is used, but clarification about what it refers to is needed.

- Line 164 and Figure 2D – what metric was used to determine "most extreme clones" and why was 7 chosen as the cutoff for number of most extreme clones used?

- Figure 2A – How was clustering done here?

Overall, the manuscript would benefit from editing for grammar and clarity.

Enc.

Reviewers' comments:

Reviewer #1 (Remarks to the Author):

The authors investigate the transcriptional and functional heterogeneity of human pre-adipocytes before, during and after adipogenic differentiation using different single cell-based analysis approaches and tools. They show that distinct subpopulations of cells exist in the subcutaneous white adipose tissue that may be associated with defined metabolic phenotypes and propose that this translates cellular heterogeneity into states of metabolic disease susceptibility. Lastly, they introduce a previously unrecognized cluster of zinc finger transcription factors that may be involved in the regulation of cell-type specific metabolic function. Overall, this study is exiting and makes use of novel technologies and integrative approaches using single cell analysis and characterization of clonally expanded pre-adipocytes. There are a few items that the authors should consider in a revised manuscript, as I am not entirely sure that all statements made in the manuscript are fully supported by the data.

Major:

- A concern is that the authors describe their results on cell heterogeneity as potentially related to disease risk yet an analysis of cells derived from metabolic disease patients is not part of the analysis. With the present data it is difficult to determine whether the heterogeneity described would indeed be related to pathology. Additional experiments with metabolically impaired patients would be extensive so this should at least be discussed more clearly. The dataset presented in this study is in itself highly interesting and of significant value to the community, even without making in more than it is.

We have revised the discussion to emphasize our results are in cells derived from healthy humans and that at this point, we have not studied this in disease states where changes might exist. These would be important in follow-up studies. We have added the following text to the discussion:

“It is worth noting that these studies, as well as ours, have thus far been limited to preadipocytes and adipose tissue derived from healthy humans. The single-cell heterogeneity of adipose in properly staged pre-disease and disease states, such as diabetes and metabolic syndrome, could reveal new mechanistic insights.”

- I would like to see a more detailed description of the single cell analysis beyond the mere mention of the analysis package that was used. Some important questions specific to the layout of the experiment described here need to be addressed, for instance how potential batch effects of experiments done on different days were tested and corrected for. Similarly, please provide a more detailed set of information on how library quality was tested and describe parameters for excluding cells which is likely in any scRNA-Seq approach. Some basic information seems to be missing, such as number of single cell libraries included/excluded in the analyses shown in main and suppl. figures.

We have added more detail to the methods section to be more explicit about the single-cell RNA seq processing methods. We have also noted the number of cells in the single-cell figures (figure 1A, figure 3B). The single-cell methods now read:

“For data analysis, we filtered out cells containing less than 1000 detected genes. On average, each cell contained ~1,600 detected genes. We used PAGODA¹⁶ to analyze the expression profiles in experiment ‘D1’ (differentiation beginning at 80% confluency, 2092 cells after QC) and experiment ‘D3’ (differentiation beginning at 100% confluency, 4319 cells after QC) (GSE53638). PAGODA finds non-redundant genes sets whose first principal components (i.e. aspects) are overdispersed. Significant aspects were chosen by $FDR \leq 0.05$. Single cell differential gene expression was performed with the package SCDE⁴⁶. As the t-SNE plots revealed two distinct clusters, we performed differential gene expression analysis between the left and right cluster in each day and in each separate experiment.”

- It would be helpful if the authors could determine in more detail whether there are any similarities between the two clusters shown in figure 1 and the clusters that potentially exist in figure 2A. Specifically, I would like to see the extent of overlaps on the level of individual differentially expressed genes between these two experiments. What is the percentage of differentially expressed genes that overlap between the two clusters? Please provide some type of Venn diagram and also lists of genes that do overlap.

We have added a Venn diagram as a new Figure 2C and moved the old figure to the supplement that shows the overlap between the differentially expressed genes between the clusters in the clone cell line data and the clusters in the day 0 (undifferentiated) cells in the single cell data from Figure 1A. We see that approximately 10% of each list is shared between the two different data sets.

- I am not sure I understand the data shown in figure S8: The distribution of gene expression across time points and differentiation strategies seems rather random and does not seem to reflect the description of the data in the figure legend or the text. This may be a misunderstanding, but it would then be helpful to clarify this. For instance, I do not see evidence for downregulation of the ZNF cluster during osteogenesis.

We apologize for the misunderstanding. Indeed, one of the conclusions supported by figure S8 is that neither osteogenesis nor chondrogenesis show a downregulation of the ZNF cluster, although individual ZNFs may show a downregulation over time in these two cell lineages. We have revised the figure captions to emphasize this point. The caption now reads:

“ZNFs may be concordantly downregulated in adipogenesis, but not in chondrogenesis or osteogenesis. RNA-seq expression profiles generated during the differentiation of fibroblasts into adipocytes, chondrocytes, or osteocytes were obtained from GEO GSE37521 (see Methods). The expression of ZNF264, ZNF490, ZNF587, and ZNF714 show lineage-specific regulation during differentiation. (A) Adipogenesis shows higher expression of the ZNF cluster in the undifferentiated state (day 0) compared to day 7 (T7). (B) Chondrogenesis suggests a time-dependent decline in ZNF490 but not other ZNFs. (C) Osteogenesis does not show time-dependent regulation of the ZNFs. Bars indicate mean \pm s.e.m. N = 2 people.”

- The data shown in S9 are not sufficient to prove enhanced or accelerated adipogenesis, especially since the data shown for the later time point somewhat contradict this conclusion. The authors should therefore conduct a comprehensive characterization of adipogenesis and metabolic function in these knockout cell lines to support their conclusion of a novel mechanism of ZNF proteins that accelerate

adipogenesis. In particular, a time course of gene expression analysis and lipid accumulation would be essential to prove the assumption of accelerated adipogenesis.

We thank the reviewer for pointing this out. After much deliberation, we agree with the reviewer that proving the mechanism of these ZNFs would require a large experimental endeavor far beyond the scope of this manuscript. As such, we have removed figure S9 and the associated data and will pursue this in more detail as follow-up to this study.

Minor

- ATXN1, a protein localized to the nucleus, is not the human homolog of murine Sca1 (a cell surface protein), it does not exist in humans.

We thank the reviewer for this correction. We have now removed the reference to ATXN1 in the text.

- Lines 212-214: I am not sure I would characterize the identification of an adipogenesis-accelerating gene as “novel”, since there is ample evidence for quite a few genes or mechanisms or reagents that enhance / accelerate adipogenesis in the literature.

We thank the reviewer for this comment and have removed the text referencing the identification of an adipogenesis-accelerating gene as novel. It is recognized however that most genetic perturbations disrupt adipogenesis, thus the search for subsystems that could accelerate this process is biologically non-trivial.

- A recent study introduced the concept of regulatory cells in adipose tissues, the A-reg, which express CD142 as surface protein (Schwalie et al, Nature 2018). Are there any links or similarities between that study and the clusters identified here?

We have examined the reported markers of A-reg as reported in Schwalie et al. for genes such as CD142 (gene F3) and ABCG1. There is some suggestive evidence that F3 may be differentially expressed between the two clusters at days 0 in experiment D1 (FDR ~ 0.44) and experiment D3 (FDR ~ 0.20), whereas ABCG1 not (consistent with the human data in Schwalie et al. where ABCG1 was a less strong markers compared to ABCG1 expression in murine preadipocytes). We have added the following text to the discussion:

“Recently, there have been several efforts to utilize single-cell sequencing to determine the cell types in human white adipose tissue³¹⁻³⁴. For example, Schwalie et al.³¹ identified Aregs, a group of anti-adipogenic stromal cells, marked by CD142 expression. Consistent with this, our data suggest CD142 (gene F3) may be differentially expressed between the day 0 clusters in the two separate differentiation experiments (FDR ~0.44 and ~0.20).”

- Lines 246-250: I am not sure I can follow the reasoning of this passage: If there is a small pool of cells that are resistant to adipogenesis, how would they be useful in a context of transient increased nutrient supply or unknown metabolic demands? Especially if there are plenty of other, non-resistant progenitors to produce adipocytes? The avoidance of adipogenesis that the authors consider useful

would contradict the physiological imperative to store nutrients effectively. Potentially related to previous comment.

We agree with the reviewer. Indeed, this is a complex interplay of theoretical possibilities. The simplest hypothesis might be that delaying differentiation might be beneficial to create a balanced population of adipocytes most consistent with a just in-time environmental response. The other speculation is that adipogenesis has an energetic cost and that adipocytes have a higher maintenance energetic demand than preadipocytes, but can store more energy. Thus, to be efficient and relatively smooth in balancing population size and energetics, adipose tissue would have the minimal number of adipocytes needed to store all the expected energy intake. The expected energy intake, however, may have a high variance. Creating adipocytes in response to a small spike in energy intake followed by a long period of low energy intake would be less efficient than simply not differentiating. Thus, to be more efficient, we would want expect adipose tissue to smooth its response to energy demands and nutrient availability to be robust to variability in both. We have revised the text such that it now reads:

“ In the context of adipose tissue biology, a small pool of preadipocytes resistant to differentiation can increase robustness to noise; for example, to avoid adipogenesis, an energetically-costly process, after a transient increase in nutrient supply or noisy sensing of pro-differentiation signals.”

- Lines 255-259: The data presented here are very exciting and could contribute to further the field significantly. I am not sure, however, that this concept is novel in the sense that it challenges “traditional thinking in disease biology”. The authors may want to reconsider such provocative statements and be more realistic, especially given that they do not explore their hypothesis experimentally and in relation to a disease and that the evidence in relation to the ZNFs they introduce has not been extensively evaluated by experiments.

The following text has been removed from the discussion to be more succinct:

“This framework may challenge traditional thinking in disease biology that attributes disease to either simple genetic or environmental factors directly leading to functional differences. The hypothesis here is that tissues may consists of cells with widely different disease risk. More has to be done to prove this hypothesis clinically.”

- Portions of the discussion seem rather unrelated to the results described in the paper. The authors could consider revising some paragraphs (for instance: lines 260-264) to make the discussion more succinct.

After revising the discussion section, the referenced text has been removed:

“We aim to catalyze the deployment of integrative network techniques to discover and validate interesting new biology. We established the potential of network techniques to identify new players and produce preliminary experimental validations for hypotheses generated by large scale network analysis which is both rare and still highly novel. We also find the potentially novel clinical background for this work highly thought provoking.”

Reviewer #2 (Remarks to the Author):

Summary and General Comments:

This manuscript describes the use of an unsupervised data generation tool, single cell RNA-seq, the development of an algorithm integrating this data with existing structured biological knowledge to generate hypotheses, and the experimental validation of these hypotheses. The study uses single cell RNA-seq (scRNA-seq) on human preadipocytes from a previous unpublished study to find two clusters of preadipocytes that did not correspond with known subtypes of these cells. These observations are supported by additional RNA-Seq and metabolic profiling is done on clonally-expanded preadipocytes. The single cell expression data is then integrated with PPI networks to identify a novel network associated with one subset of cells, and CRISPR knockdown of some of these genes supports their involvement in the differentiation of these cells. I concur with the abstract that this represents an impressive example of the use of an unsupervised experimental approach with algorithmic innovations to describe a novel mechanism in a biological system. Each step in this process requires extensive knowledge and trial and error, and it is impressive to see all steps included in one study. I recommend acceptance of this manuscript with several small changes.

Major Comments:

Some analysis of the statistical significance of the modules discovered by the network method should be included. The way this study is described, it seems like the zinc finger network may have been the clear choice for follow-up experiments, but in other studies where multiple networks could be found, an understanding of the significance of the modules would be important for prioritizing follow up. In particular, there should be more quantitative motivation for focusing on the ZNF submodule specifically. While there is a highly connected submodule of ZNFs, that reason alone does not necessarily mean that the ZNFs are any more functionally important than the other genes in the module. The authors do link the ZNFs to adipogenesis through IRF1, but that is only the fifth most enriched TF in figure S7 (and that TF ranking does not come with any measure of statistical significance) -- are any of the other TFs involved in adipogenesis, and what happens if you try to perturb IRF1? In the current form of the manuscript, focusing on ZNFs for the perturbation experiments (even if they were successful) seems a little arbitrary.

The quantitative measure used to rank different networks is the residual error of the CG decomposition. In our case, we identified up to four networks. The highest-ranking network was a ubiquitin-C network (residual error = 2582), followed by a ribosomal network (residual error = 2587), and then the ZNF network (residual error = 2593). We reasoned as the first two networks were likely capturing some known biological process in adipogenesis, whereas the ZNF network offered potentially novel biology. A jupyter notebook of these computational results were placed on our github (<https://github.com/mcrovella/CG-Demo/blob/master/CG%20Demo.ipynb>). We have edited the results section such that it now reads:

“Our method identified at least three distinct protein networks: 1) the ubiquitin-proteasome complex (residual error = 2583), 2) the ribosome (residual error = 2587), and 3) an uncharacterized protein network (residual error = 2593). As the two higher-ranking networks captured known biology in adipogenesis, we focused on the novel module consisting of 30 connected genes in the PPI (Fig 3B)²³. These genes were enriched for proteins containing KRAB domains and C2H2 zinc finger regions (Fig 3C).”

Regarding the motif analysis, the goal of this analysis was to add further credence that the ZNF network may be involved in adipogenesis, but it probably is not the only role as suggested by the enrichment of other non-adipogenic transcription factors. Indeed, when we looked at a study that performed a differentiation time course of chondrogenesis, adipogenesis, and osteogenesis (Supplementary Fig 8), the ZNFs are changing in all three mesenchymal lineages, but the data showed the four ZNFs identified in the study were all absent by day 7 in adipogenesis only, consistent with our single-cell data.

It's also possible that the ZNFs themselves are uncharacterized transcription factors. Thus, experimentally focusing on any one transcription factor (e.g. IRF1) is outside the scope of our manuscript.

- The authors really need to make clear exactly what kind of biological models and replicates are being used for both scRNA-seq and bulk experiments. How many donors? Were all cells immortalized?

We have revised the methods to include these requested details. The single-cell experiment involved primary preadipocytes derived from a single donor, whereas the bulk experiments involved immortalized preadipocytes from four donors.

- Figure S9 in the supplement is the key biological finding and should be moved to the main text. Again, the authors should also make clear what kind of replication is happening; the current description of "3 different transfections" is ambiguous. Are these all in the same cell line or different cell lines?

To answer the original question, it was the same cell line each time. However, this supplementary figure has since been removed since we have decided to do a more complete analysis of the role of the ZNFs into more complete future studies..

Minor Comments:

- Further description of the single cell RNA-seq data analysis is necessary – ex. How many cells were sequenced vs filtered out during QC? How many genes were found in each cell, on average?

We have revised the methods section on single-cell RNA seq data analysis. The section now reads:

“For data analysis, we filtered out cells containing less than 1000 detected genes. On average, each cell contained ~1,600 detected genes. We used PAGODA¹⁶ to analyze the expression profiles in experiment ‘D1’ (differentiation beginning at 80% confluency, 2092 cells after QC) and experiment ‘D3’ (differentiation beginning at 100% confluency, 4319 cells after QC) (GSE53638). PAGODA finds non-redundant genes sets whose first principal components (i.e. aspects) are overdispersed. Significant aspects were chosen by $FDR \leq 0.05$. Single cell differential gene expression was performed with the package SCDE³⁷. “

- Why are only days 0, 3, and 7 used in this study when the original study included additional time points.

We have revised the methods to point out two separate experiments were performed in the original study: one experiment (referred to experiment “D3” in original study) contained 3 time points (days 0, 3, and 7) and the other experiment (referred to experiment “D1” in the original study) contained 8 time points (days 0, 1, 2, 3, 5, 7, 9, 13). Both experiments are included in this manuscript.

For the sake of presenting our work in succinct and clear manner, each main-text figure is based on data from one of the three main experiments: figure 1 used the data from experiment “D1”, figure 2 used the data from the clonally-experiment preadipocytes, and figure 3 used the data from experiment “D3”.

- Our understanding from the previous study is that this single cell RNA-Seq data is from a single human donor. If this is true, this should be pointed out in the manuscript as it limits the generalizability of the biological findings.

Correct, in the methods we stated:

“In these data, the primary human preadipocytes were taken from commercially-available lipoaspirates (single donor; Life Technologies).”

We have revised the results such that is now explicitly states this fact:

“Primary human abdominal subcutaneous preadipocytes, derived from a single donor, were seeded...”

- Figure 2b would be more readable if the human-readable names of the GO terms were used as labels in addition to or instead of the GO accession numbers.

We have revised figure 2B to add GO names in addition the GO numbers with the associated GO name.

- Lines 161-162 describes an analysis of correlation with a phenotype. Does this refer to correlation of the individual genes in these GO lists or some sort of score on the GO terms? How was the significance of this correlation determined since different GO terms contain different numbers of genes?

The (Spearman) correlation is between a score on the GO terms and the measured phenotypes.

Briefly, PAGODA (Pathway and Gene Set Overdispersion Analysis) normalizes the gene counts in each cell accounting for sequencing depth, drop-out rate, and amplification noise. Then, the variance for each gene is renormalized based on transcriptome-wide expectations of the mean-variance relationship. The resulting matrix is then used for weighted PCA (accounting for technical noise) on a set of pre-defined or de-novo gene sets. In each gene set, the transcriptional heterogeneity is captured in the first principle component and if the variance explained is greater than expected, the gene set is overdispersed and captures some significant aspect of heterogeneity. Finally, the gene sets are clustered to reduce redundancy of those found in the same groups of cells or containing genes with similar loadings. This final matrix is a weighted PCA matrix of cells (columns) x gene set scores (rows). The gene sets in this matrix are the “aspects” commonly referred to in the manuscript, which are then correlated to the phenotype matrix (cells = columns, rows = ECAR, OCR, PPARG, GU).

In our data, the number of significant aspects was low (FDR \leq 0.05 produced 5 different aspects). The number of correlations was 20 (5 different aspects and 4 different phenotype), thus the significance is determined from 20 correlations.

- Throughout the manuscript (ex. caption to figure 2, and line 151), the word “aspects” is used, but clarification about what it refers to is needed.

We have added the following sentence to the methods under “Base-level analysis of single-cell RNA-seq”:

“PAGODA finds non-redundant genes sets whose first principal components (i.e. aspects) are overdispersed”

The term is recycled from the original PAGODA paper where an aspect can be thought of as a principal component (from PCA) that captures some biologically-meaningful variation in single-cell data. See above for a more thorough explanation.

- Line 164 and Figure 2D – what metric was used to determine “most extreme clones” and why was 7 chosen as the cutoff for number of most extreme clones used?

The most extreme clones are those from those in the furthest branches in the tree shown at the top in Figure 2B. Since we hypothesized there could be phenotypically distinct subclusters of preadipocytes within the 2 large clusters, we cut the tree at a small height to produce 5 clusters where the cluster membership ranged from 3 to 12, with the average size of 7. We have added the following text to the methods section “Total RNA isolation and RNA-seq of clonal preadipocytes”:

“To examine potential phenotypically-distinct subclusters within these two large clusters defined by gene expression, we performed hierarchical clustering and cut the tree to yield 5 clusters with average cluster size of 7 cell lines.”

- Figure 2A – How was clustering done here?

We thank the reviewer for catching this. We have added this to the methods section under “Total RNA isolation and RNA-seq of clonal preadipocytes.” The section now has the sentence:

“To estimate the number of clusters in the subjected-corrected expression profiled, we performed k-medoids clustering and selected the k that maximized the average silhouette width (k=2 to k=10).”

Overall, the manuscript would benefit from editing for grammar and clarity.

We have edited our manuscript to increase the clarity and reduce grammar inconsistencies.

REVIEWERS' COMMENTS:

Reviewer #1 (Remarks to the Author):

The authors have addressed all my comments to a sufficient degree. The study reports some highly interesting findings that will conceptually advance this field of research significantly.

Reviewer #2 (Remarks to the Author):

The authors have addressed all my requested changes.